# Application of synthesized metal-trimesic acid frameworks for the remediation of a multi-metal polluted soil and investigation of quinoa responses

Amir Zarrabi[1], Reza Ghasemi-Fasaei[1]*, Abdolmajid Ronaghi[1], Sedigheh Zeinali[2], Sedigheh Safarzadeh[1]

1 Department of Soil Science, School of Agriculture, Shiraz University, Shiraz, Iran, 2 Department of Nanochemical Engineering Faculty of Advanced Technology, Shiraz University, Shiraz, Iran

* ghasemif@shirazu.ac.ir

**Data Availability Statement:** All relevant data are within the manuscript and its Supporting Information files.

## Abstract

Metal-organic frameworks (MOFs) are structures with high surface area that can be used to remove heavy metals (HMs) efficiently from the environment. The effect of MOFs on HMs removal from contaminated soils has not been already investigated. Monometallic MOFs are easier to synthesize with high efficiency, and it is also important to compare their structures. In the present study, Zn-BTC, Cu-BTC, and Fe-BTC as three metal-trimesic acid MOFs were synthesized from the combination of zinc (Zn), copper (Cu), and iron (Fe) nitrates with benzene-1,3,5-tricarboxylic acid ($H_3BTC$) by solvothermal method. BET analysis showed that the specific surface areas of the Zn-BTC, Cu-BTC, and Fe-BTC were 502.63, 768.39 and 92.4 $m^2g^{-1}$, respectively. The synthesized MOFs were added at the rates of 0.5 and 1% by weight to the soils contaminated with 100 $mgkg^{-1}$ of Zn, nickel (Ni), lead (Pb), and cadmium (Cd). Then quinoa seeds were sown in the treated soils. According to the results, the uptakes of all four HMs by quinoa were the lowest in the Cu-BTC 1% treated pots and the lowest uptakes were observed for Pb in shoot and root (4.87 and 0.39, $\mu gpot^{-1}$, respectively). The lowest concentration of metal extracted with EDTA in the post-harvest soils was for Pb (11.86 $mgkg^{-1}$) in the Cu-BTC 1% treatment. The lowest metal pollution indices were observed after the application of Cu-BTC 1%, which were 20.29 and 11.53 for shoot and root, respectively. With equal molar ratios, highly porous and honeycomb-shaped structure, the most crystallized and the smallest constituent particle size (34.64 nm) were obtained only from the combination of Cu ions with $H_3BTC$. The lowest porosity, crystallinity, and a semi-gel like feature was found for the Fe-BTC. The synthesized Cu-BTC showed the highest capacity of stabilizing HMs, especially Pb in the soil compared to the Zn-BTC and the Fe-BTC. The highly porous characteristic of the Cu-BTC can make the application of this MOF as a suitable environmental solution for the remediation of high Pb-contaminated soils.

**Funding:** The author(s) received no specific funding for this work.

**Competing interests:** The authors have declared that no competing interests exist.

## Introduction

Remediation of HMs pollution in soil is a long-standing challenge. Because the presence of HMs ions in the surface layer of the soil leads to their bioaccumulation in plants and living organisms [1]. Human exposure to HMs such as Pb and Cd can lead to severe health consequences such as neurological disorders, cancer, and reproductive problems [2]. Unsustainable human activities and the rapid expansion of industry and agriculture has caused severe pollution of HMs, which are very dangerous for humans, environment, and ecosystems due to their high toxicity [3, 4]. Control of HMs is of higher importance compared to other pollutants such as organic pollutants due to their bioaccumulation, non-degradability, and persistence [5].

Quinoa (*Chenopodium quinoa*) is a remarkable product in terms of food security due to its highly nutritious seeds [6]. Quinoa has the genetic ability to accumulate large amounts of HMs such as chromium, Cd, and Ni in the leaf tissue [7]. Quinoa has deep and a fibrous root system that allows the plant to access nutrients in the soil solution that are unavailable to other plants. This feature increases the absorption efficiency of trace metals [8]. Due to its phytoremediation potential, quinoa absorbs HMs while prevents the transfer of these metals to the seed [9]. Quinoa seeds retain their high nutritional value even when grown in contaminated soils under irrigation with industrial wastewater [10]. The high ability of some plant roots to absorb HMs and prevent the translocation of these metals to aerial parts has been observed in previous studies. Liu et al. [11] reported that *Chlorophytum comosum* was able to absorb mercury from nutrient solution containing 800 $\mu gL^{-1}$ of this metal with total content of 0.2 $\mu gg^{-1}$ in a contaminated soil, while the transfer of this metal to aerial parts was low.

The application of amendments can change physical and chemical properties of soil and the bioavailability of metals [12]. The stabilization of HMs is mainly carried out through adsorption, complexation, reduction, and precipitation reactions, which redistribute HMs from the liquid phase to the solid phase of soil and reduce the mobility and bioavailability of these metals [13]. Conventional adsorbents have drawbacks such as high application and maintenance costs, emission of secondary pollutants, and most importantly, underperformance at low concentrations [14].

MOFs are compounds with porous structures that can be effectively used to remove HMs [15]. MOFs are a group of compounds consisting of metal ions or clusters that are coordinated with organic ligands as linkers [16]. These frameworks create three-dimensional structures and are considered as a new class of porous materials [17]. The important features of MOFs are having crystalline structure, large pores, high surface area, selective adsorption of small molecules, and controllability of their particle size and morphology [18]. MOFs are synthesized using different methods, metal ion sources, organic ligands, and solvents [19]. The metal ions used in the preparation of MOFs are Zn, Cu, Fe, Cd, magnesium, zirconium, cobalt, and aluminum [20]. Through the integration of functional groups and metal clusters, composites with diverse physical and chemical properties are obtained, which enables their application in a wide range of fields [21]. By placing metal ions in the center, MOFs can form various geometrical structures such as linear, square planar, cubic, pyramidal, triangular, tetrahedral, and octahedral [22].

$H_3BTC$ is an organic ligand with three carboxylate groups that was first introduced by Duchamp and Marsh [23] and is still an important linker for the construction of MOFs due to its predictable honeycomb-shaped crystal lattice structure and two-dimensional hydrogen bonding networks.

Some of the synthesis methods of MOFs are: room temperature synthesis, hydrothermal or solvothermal synthesis, electrochemical method, and microwave assisted synthesis. In the room temperature synthesis, MOFs are rapidly formed by mixing some metal salts and organic ligands at room temperature [24]. Hydrothermal or solvothermal is a method for synthesizing

crystals that is done under high temperature and pressure. By dissolving a metal salt in water or another solvent and transferring the solution into a steel-walled tank, the crystals are formed and grown. A temperature gradient is created between two opposite ends of the tank. At the warmer end, the metal salt is in solution, while at the cooler end, the crystals form and grow [25]. Hydrothermal synthesis is one of the new methods for producing porous materials to remediate soil pollution [26]. In the electrochemical method, an electric current is applied to anode electrode, which is immersed in a solution containing an electrolyte and an organic ligand. By applying an anodic voltage, the metal is oxidized to metal ions and released into the solution. The metal ions react with the organic ligand in the solution and a thin layer of MOF is formed on the electrode surface [27]. In microwave synthesis, microwave radiation with frequencies between 300 MHz to 300 GHz is applied to the reaction mixture to induce nucleation, which significantly shortens the crystallization time [28].

No clear study has been done so far on the effects of metal-$H_3BTC$ MOFs on the removal of HMs in contaminated soil. On the other hand, single-metal MOFs has easier synthesis steps than the synthesis of bi- or multi-metal MOFs. Due to their unique structures, MOFs have a high ability to stabilize HMs. In this study, the stabilization potential of the synthesized MOFs was investigated in the presence of a plant such as quinoa, which has a high ability to accumulate HMs. Therefore, the objectives of the present study were: i) preparation of Zn-BTC, Cu-BTC, and Fe-BTC as three metal-$H_3BTC$ MOFs and comparison of their morphology, physical, and chemical characteristics, ii) investigating the effects of the prepared metal-$H_3BTC$ MOF structures on the stabilization and bioavailability control of HMs in a multi-metal contaminated soil, and iii) investigating the responses of quinoa to the uptake of HMs after the application of the three synthesized MOFs.

## Materials and methods

### Materials

$Zn(NO_3)_2 \cdot 6H_2O$ (Zn nitrate hexahydrate), $Cu(NO_3)_2 \cdot 3H_2O$ (Copper nitrate trihydrate), $Fe(NO_3)_3 \cdot 9H_2O$ (Iron nitrate nonahydrate) and $Na_2$-EDTA (Ethylenedinitrilotetraacetic acid disodium salt dihydrate) with the purity of 99% and N, N′-dimethylformamide (DMF) with 99.8% purity (Merck). $H_3BTC$ with 95% purity (Sigma-Aldrich).

### Methods

**Soil.** A surface sample soil (depth of 0–30 cm) was selected from the study area: College of Agriculture experimental station, Shiraz, Fars province, Iran. Some physical and chemical analysis of the collected soil were as follows: pH and electrical conductivity (EC) were measured by pH-meter in saturated paste and by EC-meter in saturated extract, respectively. Soil textural class by hydrometer method [29]. The percentage of soil organic matter (OM) was measured by wet oxidation method [30]. The bioavailable fraction of HMs was extracted by 1% of $Na_2$-EDTA and measured by an atomic absorption spectrophotometer.

**$Na_2$-EDTA extraction method.** 20 mL of $Na_2$-EDTA (1%) as aqueous solution (pH = 4.8) was added to 2 g of soil, shaken for 2 h and filtered through filter paper (Whatman 42) [31].

### Synthesis of metal-$H_3BTC$ MOFs

**Zn-BTC.** 0.297 g (1 mmol) of $Zn(NO_3)_2 \cdot 6H_2O$ and 0.210 g (1 mmol) of $H_3BTC$ as a linker were dissolved in 6 mL of DMF solvent + 6 mL of ethanol + 6 mL of deionized water and this mixture was transferred into a Teflon autoclave and kept at 70°C for 4 days. After 4 days, the white crystals were separated from the solution by centrifugation at 6,000 rpm and washed with DMF [32].

**Cu-BTC.** 0.241 g (1 mmol) of $Cu(NO_3)_2 \cdot 3H_2O$ and 1 mmol of $H_3BTC$ were dissolved in 6 mL of DMF solvent + 6 mL of ethanol + 6 mL of deionized water and this mixture was transferred into a Teflon autoclave and kept at 70˚C for 24 h. After 24 h, the blue product was separated from the solution by centrifugation at 6,000 rpm and washed with DMF [32].

**Fe-BTC.** 0.342 g (1 mmol) of $Fe(NO_3)_3 \cdot 9H_2O$ and 1 mmol of $H_3BTC$ were dissolved in 20 mL of DMF solvent + 20 mL of deionized water and this mixture was transferred into a Teflon autoclave and kept at 70˚C for 24 h. After 24 h, the orange product was separated from the solution by centrifugation at 6,000 rpm and washed with DMF [33].

## MOFs characterization

The functional groups of the synthesized MOFs were determined by Fourier-transform infrared spectroscopy (FTIR) analysis (Bruker Tensor II, Germany). The appearance and size of the synthesized MOFs were recognized by field emission scanning electron microscope (FESEM) analysis. The composition of the constituent elements of the synthesized MOFs was determined by energy-dispersive X-ray spectroscopy (EDS) analysis. The crystalline state of the synthesized MOFs was analyzed by X-ray diffraction analysis using XRD, Philips model: PW1730, Cu Kα radiation, 2θ = 10–80˚. BET analysis was performed to calculate the specific surface area based on the measurement of nitrogen gas volume using vacuum degasser (BELSORP MINI II).

## Treatment of the multi-HMs contaminated soils with Zn-BTC, Cu-BTC, and Fe-BTC and plant cultivation

100 mgkg$^{-1}$ of each Zn, Ni, Pb, and Cd was added to the studied soils as nitrate sources. Soils contaminated with the HMs were incubated for one month at T = 25 ± 2˚C and field capacity moisture [34]. In order to investigate the synthesized MOFs effects on immobilizing HMs, a completely randomized design experiment was arranged including Zn-BTC, Cu-BTC, and Fe-BTC each at two rates of 0.5 and 1% of soil dry weight and each in three replications. The synthesized MOFs were added to the soils as an ultrasonicated liquid and thoroughly mixed. The soils were transferred to pots and 10 quinoa (cv. *Titicaca*) seeds were planted. A few days after germination, 5 of the healthiest seedlings were kept in each pot and the rest were weeded [7]. After eight weeks, the aerial parts of the plant were harvested and the roots were separated from the soils and the soil residues on the surface of the roots were completely washed with water. A dilute EDTA solution was used to remove HMs that may be present on the surface of the roots [35]. After washing, the aerial parts and roots were completely dried at 60˚C and then weighed and ground. The measurement of HMs in aerial parts and roots was performed by wet digestion method. The bioavailable fraction of HMs in the soils after harvesting was analyzed by extraction with $Na_2$-EDTA 1%. HMs extracted from plant parts and soils were measured by the atomic absorption spectrophotometer. After measuring the concentration of HMs, the metal uptake by shoot and root were calculated using Eqs 1 and 2 [36]:

$$\text{Shoot uptake } (\mu\text{gpot}^{-1}) = \text{shoot concentration } (\mu\text{gg}^{-1}) \times \text{shoot dry matter } (\text{gpot}^{-1}) \quad (1)$$

$$\text{Root uptake } (\mu\text{gpot}^{-1}) = \text{root concentration } (\mu\text{gg}^{-1}) \times \text{root dry matter } (\text{gpot}^{-1}) \quad (2)$$

Translocation factor (TF) between shoot and root was calculated using Eq 3. This parameter indicates the ratio of HMs quantities in the shoot to the root, according to Liu et al. [37]:

$$\text{TF} = \text{shoot metal concentration } (\text{mgkg}^{-1})/\text{root metal concentration } (\text{mgkg}^{-1}) \quad (3)$$

Metal pollution index (MPI) which shows the status of HMs in the plant was calculated using Eq 4 [38]:

$$MPI = (C_{f1} \times C_{f2} \times \ldots \times C_{fn})^{1/n} \qquad (4)$$

where, $C_{fi}$ is concentration for the metal i in the plant ($mgkg^{-1}$) and n is the total number of metals.

## Wet digestion

0.1 g of dry matter of the plant was digested in 15 mL of one to three (V/V) mixture of HCl/HNO$_3$ at 180˚C for 1 h on an electric heating plate. After cooling, 2 mL of 30% hydrogen peroxide was added and heated for another 20 minutes at 150˚C, and this procedure was repeated two more times. After the digestion process, the sample was diluted with HNO$_3$ (10 m L/L) to a final volume of 50 mL. Finally, the concentration of HMs in the digestion solution was measured with the atomic absorption spectrophotometer [39].

## Statistical analysis

The control treatment plus three of MOF treatments were each in two rates with codes 1 to 7, which were numbered in the software as 7 codes 1 to 3 (three replicates). Compare means analysis as Duncan's multiple range test ($P < 0.05$) (One-Way ANOVA) were done using IBM SPSS 25. Drawing of graphs and figures was performed by Excel software package.

## Results and discussion

### Soil

Some physical and chemical characteristics of the studied soil are shown in Table 1. According to the results, the soil pH was in the neutral range (7.3). The soil contained very low concentrations of bioavailable Zn, Ni, Pb, and Cd, which were insignificant, 0.25, 0.18, and 1 $mgkg^{-1}$, respectively. With 15.4% of clay, 30% of sand, and 54.6% of silt, the soil texture class was identified as Silt Loam. The percentage of soil OM was very low (0.98%). According to the low percentages of clay and OM, it can be concluded that the studied soil did not have strong adsorption capacity.

### FTIR analysis

The FTIR results of the synthesized Zn-BTC, Cu-BTC, and Fe-BTC are shown in Fig 1. The number of observed peaks from the highest to the lowest were 25 (455.17 to 3430.88 cm$^{-1}$), 23 (441.72 to 34.44.74 cm$^{-1}$), and 19 (489.19 to 3387.10 cm$^{-1}$) for Fe-BTC, Zn-BTC, and Cu-BTC, respectively. According to these results, the strongest and the most abundant peaks were observed for the Fe-BTC. For all three MOFs, the observed peaks of nearly 500 cm$^{-1}$ confirmed

**Table 1. Some physical and chemical characteristics of the studied soil.**

| pH | EC (dSm$^{-1}$) | Clay | Sand | OM | Zn-EDTA | Ni-EDTA | Pb-EDTA | Cd-EDTA |
|---|---|---|---|---|---|---|---|---|
| | | . . . . . . . . % . . . . . . . | | | . . . . . . . . . . . . . . . . . . . . . . . . . . . . mgkg$^{-1}$ . . . . . . . . . . . . . . . . . . . . . . . . | | | |
| 7.3 | 2.3 | 15.4 | 30 | 0.98 | ND | 0.25 | 0.18 | 1 |

**EC:** Electrical conductivity

**OM:** Organic matter

**ND:** Not detected

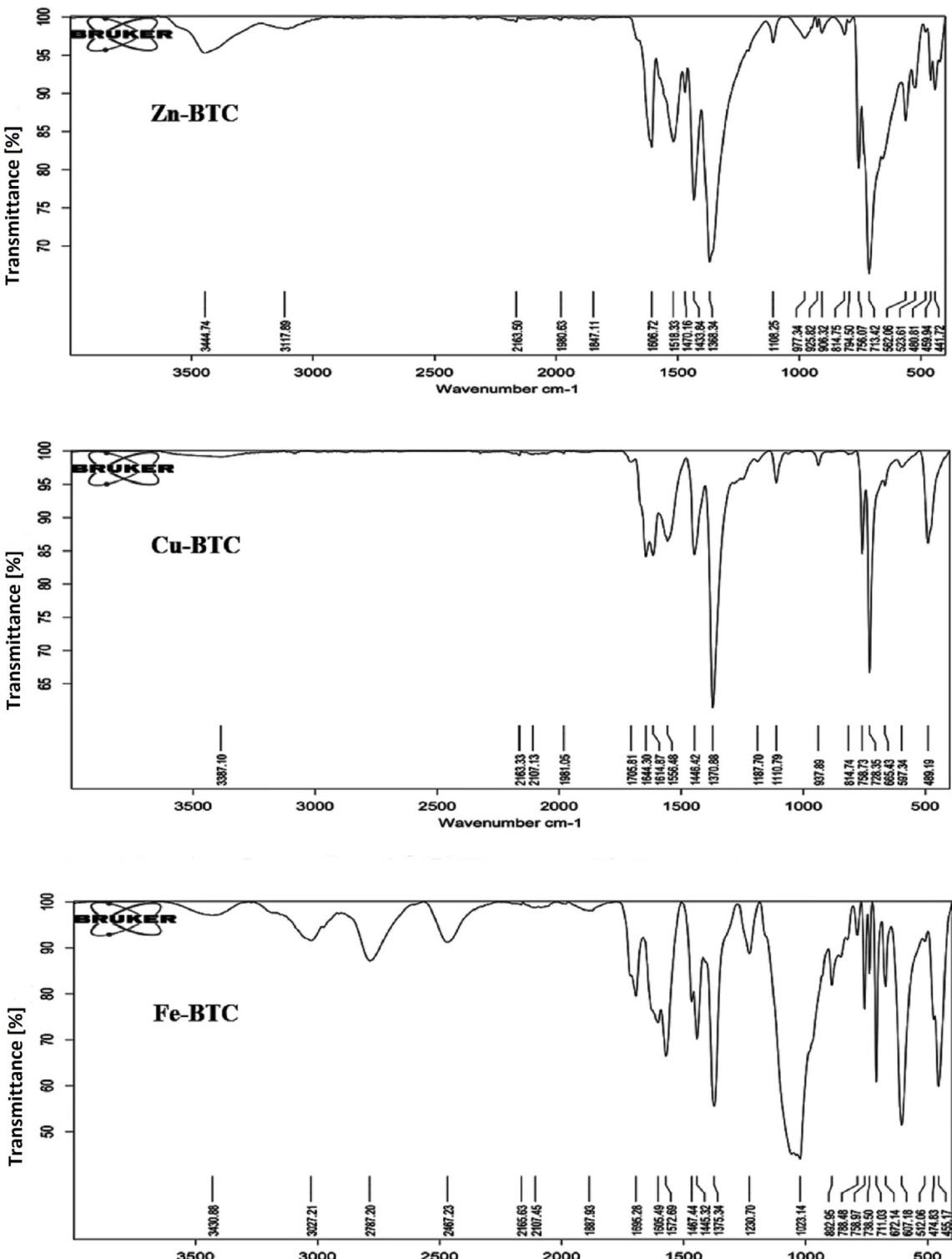

**Fig 1. Infrared spectra of the synthesized Zn-BTC, Cu-BTC, and Fe-BTC.**

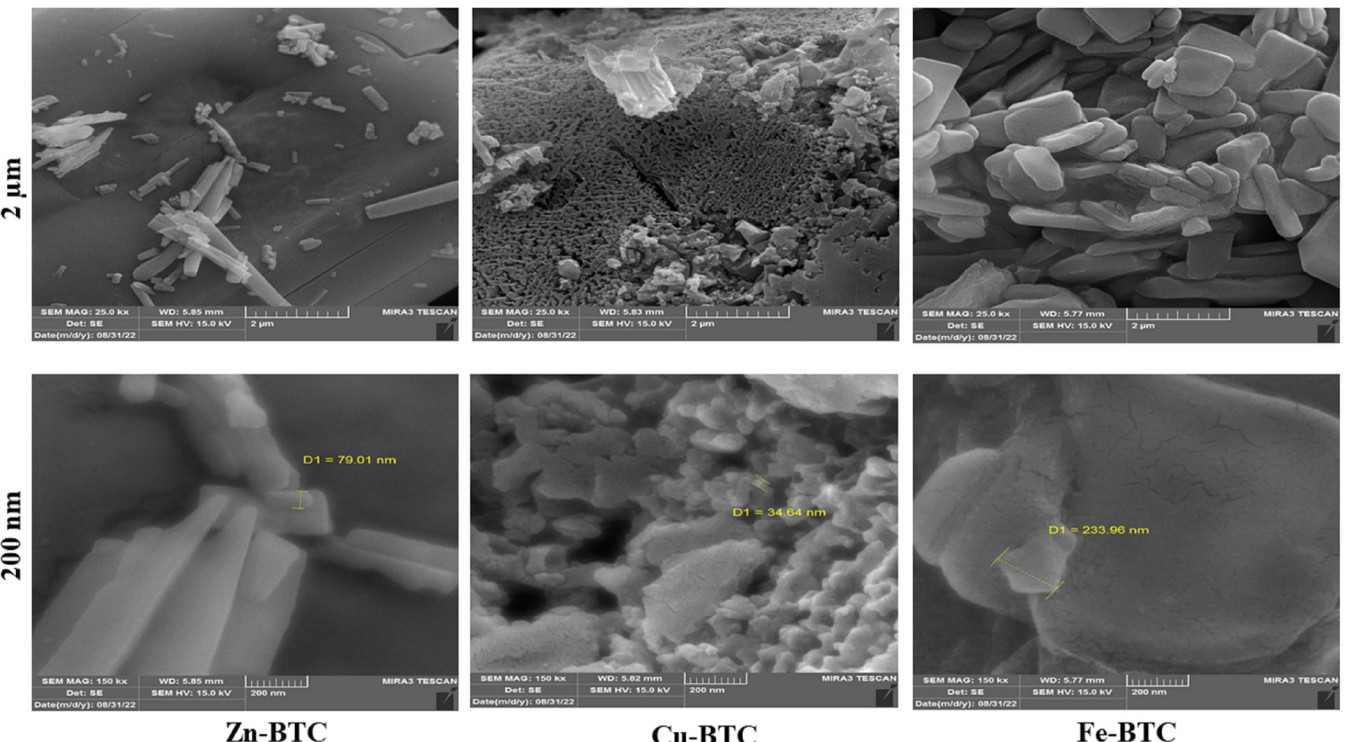

**Fig 2. FESEM graphs of the synthesized Zn-BTC, Cu-BTC, and Fe-BTC (Exterior and interior views).**

the presence of metal-oxygen bonds [40, 41]. This peak was stronger in the Fe-BTC. For the Zn-BTC, the strongest observed peaks were observed at the wave numbers of 713.42 and 1368.34 $cm^{-1}$, which represented stretch alkenes and aromatics (C-H). 1606.72 $cm^{-1}$ indicated amide groups (C = O). For the Cu-BTC, similar important peaks were observed in two wave numbers of 728.35 and 1370.88 $cm^{-1}$ which confirmed the same functional groups which were observed for Zn-BTC. In addition, for Cu-BTC, the bifurcated peaks with less sharpness at 1705.81 and 1644.30 $cm^{-1}$ indicated carbon and oxygen double bonds of ketone and amide groups, respectively. The FTIR analysis showed more sharp peaks for Fe-BTC than the other two MOFs. The peak at 1023.14 $cm^{-1}$ showed alcohols, ethers, esters, carboxylic acids, anhydrides groups (C-O), and that of 607.18 $cm^{-1}$ represented bromide groups (C-X). The vibration of 1375.34 $cm^{-1}$ confirmed the presence of bend alkanes (-$CH_3$) and the wave number of 711.03 $cm^{-1}$ indicated stretch alkenes and aromatics (C-H) groups [42].

## SEM and EDS analysis

In the present study, Zn-BTC, Cu-BTC, and Fe-BTC were synthesized by solvothermal method in white, blue, and orange powders, respectively. The synthesized Fe-BTC had a semi-gel state, which turned into a powder after air drying. FESEM images of the three synthesized MOFs at two magnifications of 2 μm and 200 nm are shown in Fig 2. Zn-BTC, Cu-BTC, and Fe-BTC have been synthesized in diverse appearance forms. Zn-BTC was in the form of interwoven rod-shaped objects, Cu-BTC with high crystallinity, porous, and honeycomb structure, and finally the square flake-shaped particles were observed for the synthesized Fe-BTC. In previous studies, rod-shaped Zn-BTC has been synthesized by Mostaanzadeh et al. [43] and Wang et al. [44]. Porous crystalline Cu-BTC synthesized by Iqbal et al. [45]. Also for Fe-BTC,

**Table 2. Elemental composition of the synthesized MOFs through EDS analysis.**

| | Elements (Weight %) | | | | |
|---|---|---|---|---|---|
| | **Zn** | **Cu** | **Fe** | **O** | **C** |
| **Zn-BTC** | 15.1 | - | - | 35.44 | 49.46 |
| **Cu-BTC** | - | 14.72 | - | 40.52 | 44.76 |
| **Fe-BTC** | - | - | 12.95 | 50.54 | 36.51 |

flake-like morphology has been synthesized by Salazar-Aguilar et al. [46]. The cross section of the synthesized rods for the Zn-BTC was 79.01 nm. The smallest particles forming the network structure in the Cu-BTC were 34.64 nm and the smallest size for the flakes forming the Fe-BTC was shown at 233.96 nm.

Table 2 shows the results of EDS analysis. The chemical composition of the synthesized MOFs based on weight percentage was as follows: 15.1%, 35.44%, and 49.46% of Zn, O, and C, respectively for the Zn-BTC. The percentages of 14.72, 40.52, and 44.76 for Cu, O, and C, respectively were for the Cu-BTC and the chemical composition of the Fe-BTC included 12.95%, 50.54%, and 36.51% of Fe, O, and C, respectively.

## BET analysis

The results of BET analysis of the synthesized MOFs are shown in Table 3. Due to its very porous and crystalline structure, the Cu-BTC had the highest specific surface area of 768.39 ($m^2g^{-1}$) and pore volume of 1.28 ($cm^3g^{-1}$). The specific surface area of 708 ($m^2g^{-1}$) for Cu-BTC was reported in previous studies by Siew et al. [47]. Al-Janabi et al. [48] reported a specific surface area between 700 and 1000 ($m^2g^{-1}$) for their synthesized Cu-BTCs. The specific surface area of Zn-BTC synthesized in this study was 502.63 ($m^2g^{-1}$) and its pore volume was 0.66 ($cm^3g^{-1}$). Due to the shape of an intertwined rods with a width of about 80 nm (Fig 2), the produced Zn-BTC had a significant specific surface area. This result was in good agreement with Aftab et al. [49] who synthesized rod-shaped Zn-BTC with rigid porosity and specific surface area of 545.28 ($m^2g^{-1}$). The lowest specific surface area (92.4 $m^2g^{-1}$) and pore volume (0.11 $cm^3g^{-1}$) was found for the Fe-BTC. The lower specific surface area of the Fe-BTC was due to the fact that these MOFs were synthesized in flake forms with almost disordered nature. Therefore, more external surfaces and low interior porosity of the Fe-BTC caused inaccessibility for $N_2$ adsorption [33]. The small pore volume of Fe-BTC creates pocket-type pores that are inaccessible to the $N_2$ molecules [50]. The amorphous form and low crystallinity for Fe-BTC have been observed in many previous studies. An amorphous and non-porous structure for Fe-BTC with a low BET surface area (17 $m^2g^{-1}$) and pore volume (0.0232 $cm^3g^{-1}$) was synthesized by Oveisi et al. [51].

## XRD analysis

The XRD diffraction pattern of the synthesized MOFs is shown in Fig 3. The presence of sharp peaks in 2θ values of 18.74˚, 17.64˚, 21.94˚ and 27.19˚, respectively, for Zn-BTC confirmed its crystalline nature. The presence of peak at 17.8˚ for Zn-BTC was reported in previous studies by Bhardwaj et al. [52]. In addition, weaker peaks between 20˚ and 30˚ have been reported by

**Table 3. BET surface area for the synthesized MOFs.**

| | **Zn-BTC** | **Cu-BTC** | **Fe-BTC** |
|---|---|---|---|
| $a_{s,BET}$ ($m^2g^{-1}$) | 502.63 | 768.39 | 92.4 |
| pore volume ($cm^3g^{-1}$) | 0.66 | 1.28 | 0.11 |

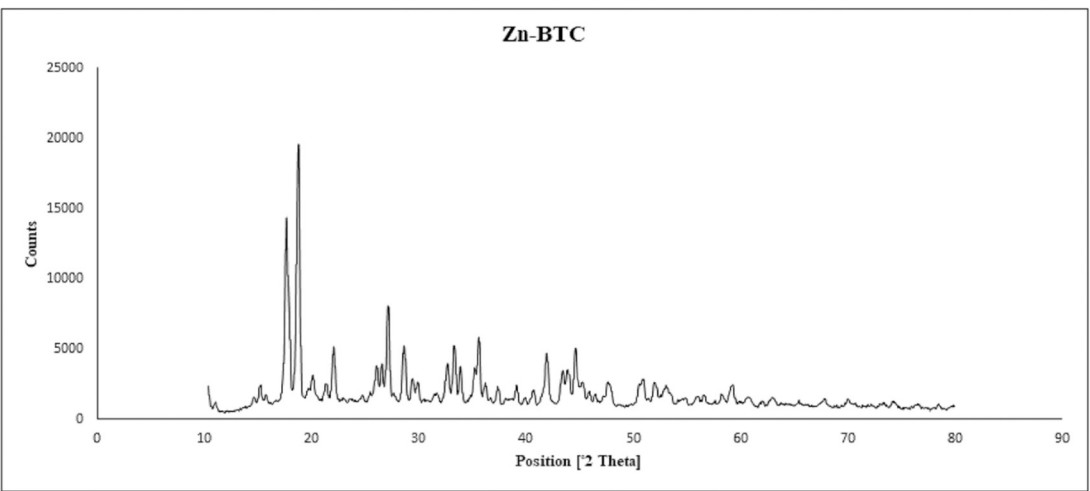

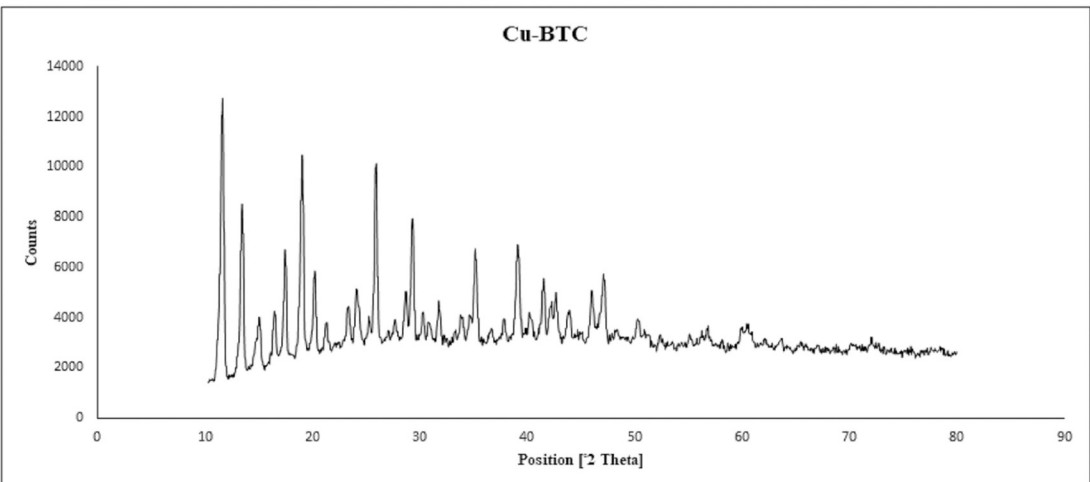

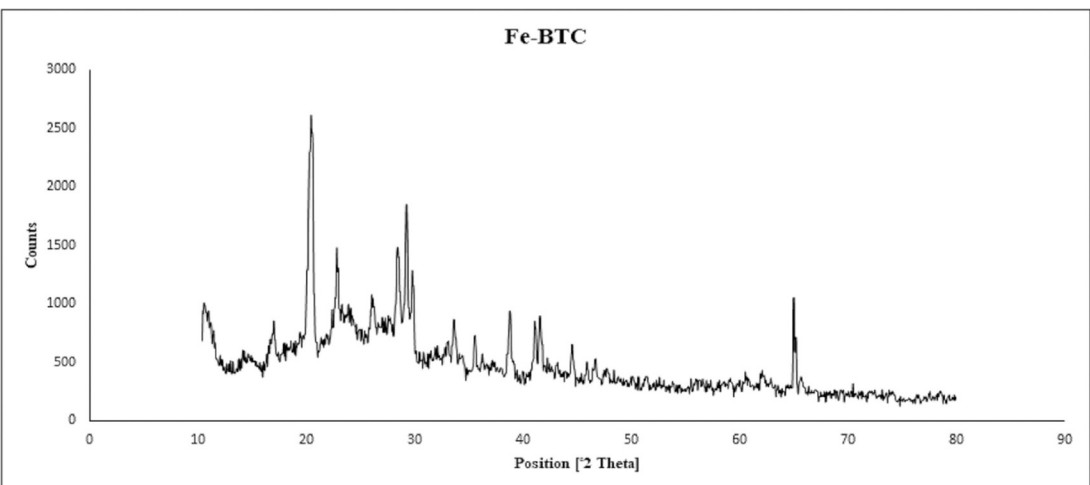

**Fig 3. XRD analysis of the synthesized Zn-BTC, Cu-BTC, and Fe-BTC.**

Aftab et al. [49] as indicating the crystalline structure of Zn-BTC. XRD analysis showed that the most regular sharp peaks were observed for the synthesized Cu-BTC, which consisted of 11.69°, 19.09°, 25.99° and 13.44° in an order from the highest to lowest. The peak of 11.7° was reported as indicating the Cu-BTC structure by Siew et al. [45]. According to Fig 3, the intensity of the peaks observed for the Cu-BTC has decreased regularly in a stepwise manner with the increase of 2θ. This result indicated the high regularity of the Cu-BTC crystalline structure. According to Bunaciu et al. [53], as the average crystal size decreases, the angle between the radiated ray and its reflection increases and its intensity decreases. Therefore, the decrease in the intensity of the Cu-BTC peaks with the increase of 2θ means that the radiation first hit the coarse crystals and gradually hit the smaller crystals. XRD analysis showed that the peaks observed for the synthesized Fe-BTC were much more irregular than the other two MOFs, which were 20.39°, 29.19° and 64.99°. The nature of the Fe-BTC was different from other two MOFs and it was synthesized in a semi-gel like form. In addition, the Fe-BTC due to having a flake shape with a low specific surface area that was mentioned earlier, had a semi-amorphous structure. Low crystalline, gel-like featured, short-range ordered Fe-BTCs have been synthesized previously by Autie-Castro et al. [50] and Kitagawa et al. [54].

## Effects of the synthesized Zn-BTC, Cu-BTC, and Fe-BTC on the HMs uptake by quinoa

The effects of the synthesized Zn-BTC, Cu-BTC, and Fe-BTC and their rates on the uptake of HMs by quinoa shoot are shown in Fig 4. According to the data presented, in the control treatment (without the synthesized MOFs), the HMs uptakes by quinoa shoot from the highest to lowest were for Zn, Pb, Cd, and Ni which were 181.43, 188.1, 166.49, and 150.33 μgpot$^{-1}$, respectively. The difference between the rates of 0.5 and 1% in the Cu-BTC treatments for all four HMs was insignificant, and only at the rates of 1%, the uptake of the HMs by plant shoot was lower compared to the rates of 0.5%. Application of 0.5 and 1% of Zn-BTC, Cu-BTC, and Fe-BTC significantly reduced the uptake of Zn by the plant shoot compared to the control, and this effect was stronger in the Cu-BTC-treated soils. The lowest uptakes of Zn were observed after the addition of Cu-BTC at 1 and 0.5%, which were 8.99 and 14.19 μgpot$^{-1}$, respectively followed by 39.44 and 30.28 μgpot$^{-1}$ which occurred after the application of 0.5 and 1% of Zn-BTC, respectively. The addition of Fe-BTC rates also significantly reduced Zn uptake by the plant in comparison with the control. The lowest uptake of Ni by quinoa shoot was 8.97 μgpot$^{-1}$, which was observed after the addition of Cu-BTC 1% and was significantly different from the control. Also, with the addition of Cu-BTC 0.5%, the Ni in shoot was 14.94 μgpot$^{-1}$. By adding the rates of 1 and 0.5% of Zn-BTC, the Ni uptake by shoot were 11.57 and 23.36 μgpot$^{-1}$, respectively and both had a significant difference compared to the control. The Ni uptakes by shoot were 65.06 and 61.27 μgpot$^{-1}$ after the application of 1 and 0.5% of the Fe-BTC, respectively, which were significantly different from the control treatment. The lowest Pb uptake by shoot was 4.87 μgpot$^{-1}$, which was observed after adding 1% of Cu-BTC followed by 7.86 μgpot$^{-1}$ after the application of Cu-BTC 0.5%. Significant differences with the control were observed after adding Zn-BTC and Fe-BTC 1%, which were 32.39 and 83.58 μgpot$^{-1}$, respectively. The application of the Zn-BTC, Cu-BTC, and Fe-BTC and their rates significantly reduced the shoot uptake of Cd compared to the control. The lowest uptakes of Cd were observed in the Cu-BTC 0.5 and 1% treatments, which were 6.46 and 9.93 μgpot$^{-1}$, respectively. According to the results, it was found that the potential of the synthesized MOFs to prevent the entry of HMs into the quinoa shoot was as follows: Cu-BTC > Zn-BTC > Fe-BTC. In addition, the highest efficiency of the Cu-BTC to preclude HMs uptakes was observed for Pb followed by Cd.

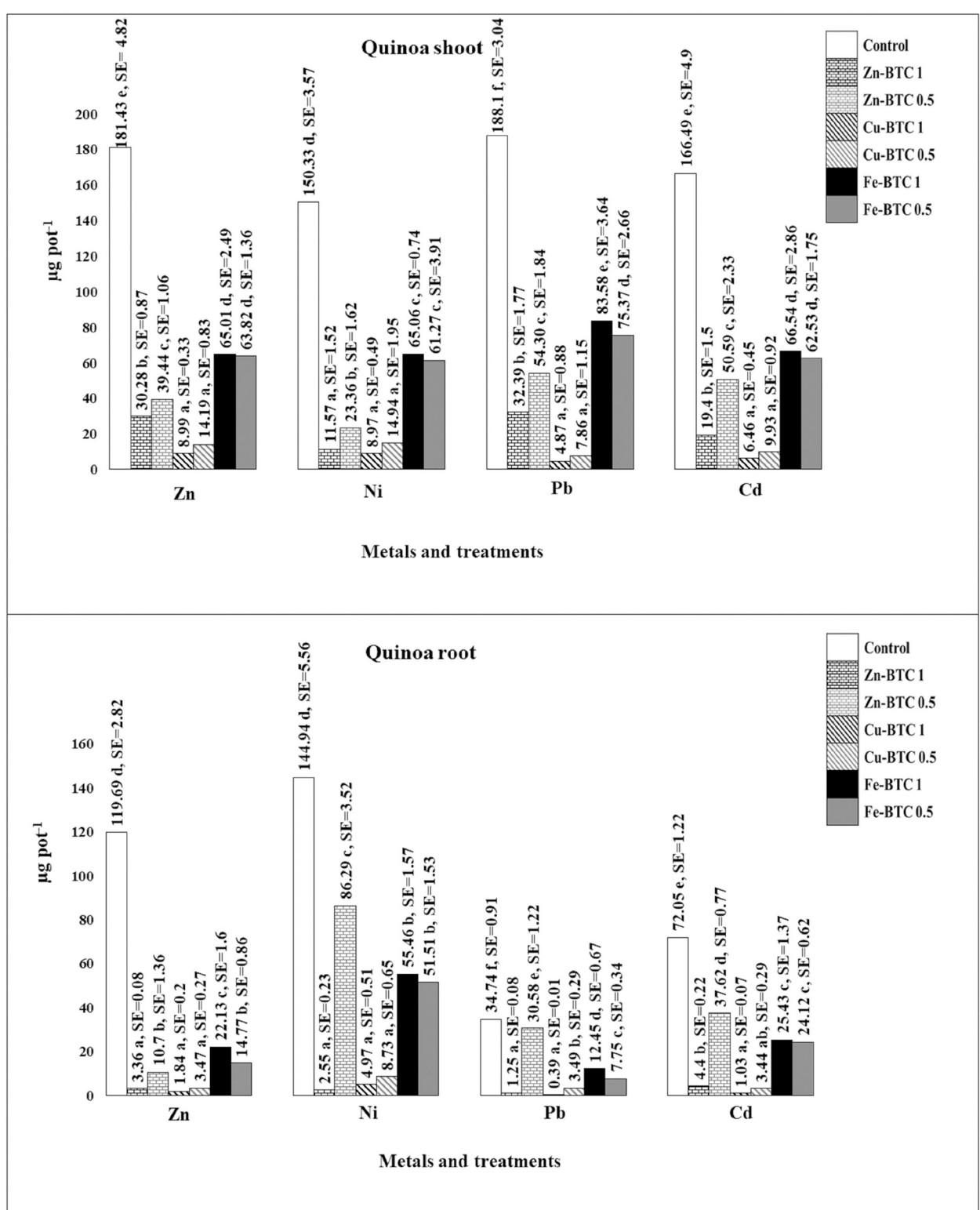

**Fig 4. Effects of Zn-BTC, Cu-BTC, and Fe-BTC on the uptake of HMs by quinoa shoot and root (µgpot⁻¹).** Compare mean analysis have been done for each metal separately. Values with the same letters indicate non-significant difference at P < 0.05 according to the Duncan's multiple range test. Each value is the average of three replications. SE: Std. Error.

The effects of the synthesized Zn-BTC, Cu-BTC, and Fe-BTC and their rates on the uptake of HMs by quinoa root are presented in Fig 4. All three synthesized MOFs caused significant reductions on the uptake of Zn, Ni, Pb, and Cd by plant roots as compared to the control. The uptakes of HMs by plant roots without the addition of the synthesized MOFs (control) from the highest to the lowest were 144.94, 119.69, 72.05, and 34.74 µgpot$^{-1}$ for Ni, Zn, Cd, and Pb, respectively. By adding the rates of 1 and 0.5% of Zn-BTC, the uptake of Zn by root decreased significantly from 119.69 to 3.36 and 10.7 µgpot$^{-1}$, respectively. The lowest uptake of Zn was 1.84 µgpot$^{-1}$, which was observed following the application of Cu-BTC 1%. Application of Cu-BTC 0.5% reduced Zn uptake to 3.47 µgpot$^{-1}$. Zn root uptake values after the application of the Fe-BTC at the rates of 1 and 0.5% were 22.13 and 14.77 µgpot$^{-1}$, respectively. The rates of 1 and 0.5% of Zn-BTC had a significant difference in reducing root Ni uptake. The value of Ni uptake in Zn-BTC 1% treatment was 2.55 µgpot$^{-1}$ while in Zn-BTC 0.5% was 86.29 µgpot$^{-1}$. The addition of Cu-BTC significantly reduced root Ni uptakes, as their values from 144.94 (control) reached 4.97 and 8.73 µgpot$^{-1}$ in the treatments of Cu-BTC 1 and 0.5%, respectively. Following the application of 1 and 0.5% of Fe-BTC, Ni uptake decreased from 144.94 to 55.46 and 51.51 µgpot$^{-1}$, respectively. The addition of 1 and 0.5% of the Zn-BTC decreased root Pb uptake from 34.74 to 1.25 and 30.58 µgpot$^{-1}$, respectively, and a significant difference was also observed between these two treatments. Root Pb uptake drastically decreased following the addition of 1 and 0.5% of Cu-BTC to 0.39 and 3.49 µgpot$^{-1}$, respectively. A significant reduction of Pb from 34.74 to 12.45 and 7.75 µgpot$^{-1}$ was observed with the addition of 1 and 0.5% of the Fe-BTC, respectively. The treatment of Zn-BTC 1% reduced root Cd uptake much more and with a significant difference compared to the rate of 0.5%. Root Cd uptake under the effect of the Cu-BTC 1 and 0.5% significantly decreased from 72.05 to 1.03 and 3.44 µgpot$^{-1}$, respectively. The values of 24.12 and 25.43 µgpot$^{-1}$ of Cd were observed following the application of 1 and 0.5% of the Fe-BTC, respectively, which had a significant difference with the control. According to the results, the highest reduction in the uptake of HMs by quinoa roots was observed under the influence of Zn-BTC 1% and Cu-BTC 1% followed by Cu-BTC 0.5%. Finally, the highest uptake of HMs by the roots happened under the effect of the Fe-BTC rates. No significant difference was observed in the uptake of all four HMs by plant root and shoot between 1 and 5% of the Cu-BTC treatments. Therefore, it can be sufficiently applied the Cu-BTC 0.5% to control the uptake of HMs by the studied plant.

The high differences in the uptakes of HMs by roots between the rates of 1 and 0.5% of Zn-BTC were most probably due to the high stunted quinoa growth in the presence of 1% of the Zn-BTC. In addition, the Cu-BTC 1 and 0.5% probably caused a decrease in root growth and significantly reduced HMs uptake values. However, stunted root growth resulted from the addition of Fe-BTC rates was less than the other two MOFs treatments, and the decrease in root growth under the influence of this MOF was much lower than the effects of Zn-BTC and Cu-BTC treatments. Such a reduction was also observed in the growth of shoots, but it was less than that of the roots. According to Table 4, the dry weights of shoot and root in the control were 4.02 and 0.72 gpot$^{-1}$, respectively. Shoot dry weight decreased to 1.12 and 0.98 gpot$^{-1}$ in Zn-BTC 1% and 0.5% treatments, respectively and root dry weight decreased to 0.52 and 0.17 gpot$^{-1}$ in 1 and 0.5% of Zn-BTC, respectively. Under the influence of the Cu-BTC 1 and 0.5%, shoot dry weight reduced to 0.47 and 0.37 g pot$^{-1}$, and root dry weight reduced to 0.18 and

**Table 4. Effect of the synthesized MOFs rates on the dry weights of quinoa root and shoot (each value is the average of three replicates).**

| Treatments | Control | Zn-BTC (1%) | Zn-BTC (0.5%) | Cu-BTC (1%) | Cu-BTC (0.5%) | Fe-BTC (1%) | Fe-BTC (0.5%) |
|---|---|---|---|---|---|---|---|
| Shoot dry weights (gpot$^{-1}$) | 4.02 | 0.98 | 1.12 | 0.35 | 0.47 | 1.89 | 1.64 |
| Root dry weights (gpot$^{-1}$) | 0.77 | 0.17 | 0.52 | 0.12 | 0.18 | 0.58 | 0.32 |

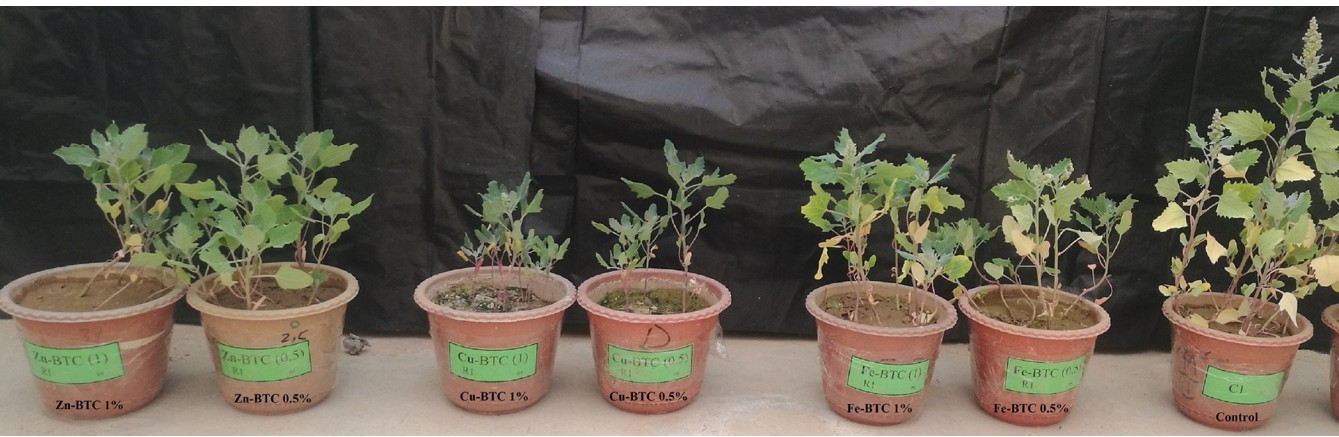

**Fig 5. Effects of the synthesized MOFs rates on vegetative growth of quinoa.**

0.12 gpot$^{-1}$, respectively. According to these observations, the 1% rate of Zn-BTC and Cu-BTC decreased quinoa shoot growth compared to the rate of 0.5% of these two MOFs. This growth reduction in the root was also similar to the shoot and was observed much more in the root growth. The effect of Fe-BTC on shoot and root growth was different compared to the other two MOFs. The shoot dry weight increased from 1.64 gpot$^{-1}$ at the rate of 0.5% to 1.89 gpot$^{-1}$ at the 1% rate of Fe-BTC. The root dry weight also increased from 0.32 to 0.58 gpot$^{-1}$ from 1% to 0.5% of the Fe-BTC.

The effects of three synthesized MOFs rates on the vegetative growth of quinoa is shown in Fig 5. The highest vegetative growth was observed for the control and all three synthesized MOFs caused a decrease in the vegetative growth of the plant compared to the control. The lowest plant growth was observed under the effect of Cu-BTC 1% followed by Cu-BTC 0.5%. Compared to the Cu-BTC treatments, vegetative growth order under the effect of Zn-BTC was as follows: Zn-BTC 0.5% > Zn-BTC 1%. Among the three MOFs, the highest vegetative growth occurred under the effect of the Fe-BTC treatments and its order was as follows: Fe-BTC 1% > Fe-BTC 0.5%. According to vegetative growth orders, in the Zn-BTC and Cu-BTC treatments, the metal uptake by shoot and root (Eqs 1 and 2) was lower in 1% rate than 0.5%. While the uptakes in the Fe-BTC treatments at the rate of 1% was higher than 0.5% (Fig 4). As shown in Fig 5, it is clear that the plants treated with the Cu-BTC were completely green, while they had the lowest vegetative growth, and the plants treated with the Zn-BTC had a brighter green color and higher vegetative growth. Since there were no effects of MOFs in the control treatment, vegetative growth was the highest, but paleness and chlorosis were observed in the leaves. Similar visual symptoms were observed in the pots under the Fe-BTC treatments, which was due to lower stabilizing of HMs in the soil and subsequently higher uptake of these metals by the plant. According to previous researches the bioaccumulation of HMs in plant causes paleness in the leaves due to disrupting the photosynthesis process [55–58].

It seems that the HMs uptake behavior of quinoa and the vegetative growth of this plant when affected by Zn-BTC, Cu-BTC, and Fe-BTC treatments were related to the structural characteristics of these MOFs. Previous results showed that the synthesized Cu-BTC had the smallest crystal size, highly porous and honeycomb structure, and the highest specific surface area. Accordingly, this MOF immobilized HMs in the soil more strongly and prevented their uptake by the plant and in the other hand caused the greatest reduction in growth. Higher reduction in HMs uptake in the presence of the Cu-BTC was not only due to the higher metal immobilization but also because of lower dry matter in this treatment. The synthesized Zn-

BTC had larger crystal size and lower specific surface than the Cu-BTC. Accordingly, the plants under the effect of the Zn-BTC absorbed more HMs than those treated with the Cu-BTC and had higher vegetative growth. The synthesized Fe-BTC had the largest size, semi-crystalline structure and the lowest specific surface area and the lowest HMs stabilizing capacity. Therefore, this MOF caused the highest HMs uptake by plant. The Fe-BTC caused the lowest stunted growth resulted in the highest biomass production as compared to Zn-BTC and Cu-BTC treatments. According to Lelouche et al. [59], MOFs synthesized by $H_3BTC$ linkers due to the presence of benzene ring and carboxyl groups have strong antimicrobial properties. On the other hand, MOFs have crystalline structure and high chemical, thermal, and mechanical stability [60]. Therefore, it can be resulted that these structures are not easily disintegrated in soil. It is likely that the stunted growth of quinoa in the presence of MOFs was due to some harmful effects that these materials caused following their entrance into the plant. Meanwhile, the Cu-BTC, due to having the smallest size, entered the plant in a larger quantity and caused the highest growth reduction. Liang et al. [61] reported the presence of terbium MOFs $[Tb_2(BDC)_3(H_2O)_4]$ in the xylem and phloem of *Syngonium podophyllum*. They found that the existence of the MOFs in the water transport channels prevented water and nutrients transport in the plant.

## Concentrations of HMs in the plant, TF, and MPI

The concentrations of Zn, Ni, Pb, and Cd in the shoot and root of quinoa and their TF are shown in Table 5. The highest concentrations of all four metals was observed for both shoot and root in the control treatment. The lowest concentrations for all four metals in shoot and root were observed in the treatment of Cu-BTC 1% followed by Cu-BTC 0.5%, respectively. The lowest metal concentrations were for Pb in shoot and root, which where 13.91 and 3.25 $mgkg^{-1}$, respectively. These results were similar to those observed for metal uptakes (Fig 4). The metal concentrations in the control treatment for Zn, Ni, and Cd was higher in root than in shoot, and the highest concentration in the root was found for Ni (188.23 $mgkg^{-1}$). In the present study, the lowest TF values for Zn (0.29), Ni (0.20), and Cd (0.44) were observed in the control treatment (without the application of MOFs). As a result, quinoa was able to prevent the entry of Zn, Cd, and especially Ni into its shoot. The lower TF in the control treatment can be attributed to the higher absorption of metals by root due to lower negative effects of MOFs and higher root dry weight compared to other treatments. The remarkable result was the lowest TF values (0.13 to 0.79 among seven treatments) for Ni. These values showed the high tendency of quinoa to accumulate Ni in the root and very low transfer of this metal to shoot. TF

**Table 5. Concentrations of HMs in quinoa shoot and root ($mgkg^{-1}$) and TF.**

| Treatments | Zn | | | Ni | | | Pb | | | Cd | | |
|---|---|---|---|---|---|---|---|---|---|---|---|---|
| | Shoot C | Root C | TF | Shoot C | Root C | TF | Shoot C | Root C | TF | Shoot C | Root C | TF |
| Control | 45.13 | 155.44 | 0.29 | 37.40 | 188.23 | 0.20 | 46.79 | 45.12 | 1.04 | 41.42 | 93.57 | 0.44 |
| Zn-BTC (1%) | 30.96 | 19.76 | 2.88 | 11.81 | 15.00 | 0.79 | 33.10 | 7.35 | 4.50 | 19.80 | 25.9 | 0.76 |
| Zn-BTC (0.5%) | 35.29 | 20.58 | 3.03 | 20.86 | 165.94 | 0.13 | 48.54 | 58.81 | 0.83 | 45.18 | 72.35 | 0.62 |
| Cu-BTC (1%) | 25.71 | 15.33 | 1.68 | 25.64 | 41.42 | 0.62 | 13.91 | 3.25 | 4.28 | 18.51 | 8.58 | 2.46 |
| Cu-BTC (0.5%) | 30.25 | 19.28 | 1.57 | 31.83 | 48.50 | 0.66 | 16.72 | 19.39 | 0.86 | 21.14 | 19.11 | 0.86 |
| Fe-BTC (1%) | 34.45 | 38.11 | 0.90 | 34.37 | 95.57 | 0.36 | 44.22 | 21.41 | 2.07 | 35.15 | 43.79 | 0.80 |
| Fe-BTC (0.5%) | 38.97 | 46.11 | 0.85 | 37.36 | 160.91 | 0.23 | 46.01 | 24.16 | 1.90 | 38.13 | 75.32 | 0.51 |

C: concentration, TF: translocation factor.

Each value is the average of three replicates

**Table 6. Metal pollution index (MPI) of quinoa treated by the synthesized MOFs (each value is the average of three replicates).**

|  | Control | Zn-BTC (1%) | Zn-BTC (0.5%) | Cu-BTC (1%) | Cu-BTC (0.5%) | Fe-BTC (1%) | Fe-BTC (0.5%) |
|---|---|---|---|---|---|---|---|
| **Quinoa shoot** | 42.52 | 22.12 | 35.64 | 20.29 | 24.15 | 36.83 | 39.97 |
| **Quinoa root** | 105.42 | 15.41 | 61.64 | 11.53 | 24.26 | 42.98 | 60.61 |

values for Cd have been reported for quinoa in previous studies. Bamagoos et al. [62] found that with the addition of 100 µM Cd in a hydroponic culture medium, the TF of this metal (between shoot and root) for quinoa was 0.9.

MPI values for the studied plant are shown in Table 6. The highest MPI index was 105.42 in root of the control treatment. This high value can be mostly attributed to the high uptake of Ni by the roots. After adding MOFs, the MPI index decreased in both shoot and root. The lowest MPI values were observed after the application Cu-BTC 1%, which were 20.29 and 11.53 for shoot and root, respectively. After Cu-BTC, the lowest MPI indices were observed following the addition of the Zn-BTC 1% (22.12 and 15.41for shoot and root, respectively). MPI results confirmed that the Cu-BTC had the highest immobilization capacity for HMs.

## Effects of the synthesized Zn-BTC, Cu-BTC, and Fe-BTC on EDTA-extractable HMs in the post-harvest soils

Data presented in Fig 6. shows the effects of the three synthesized MOFs on HMs extracted by EDTA. The concentration of HMs extracted with EDTA in the control from the highest to the lowest were for Cd, Pb, Zn, and Ni, including 62.66, 60.8, 56.8, and 42.73 mgkg$^{-1}$, respectively. The highest Zn concentration was observed under the effect of the Zn-BTC 1% treatment,

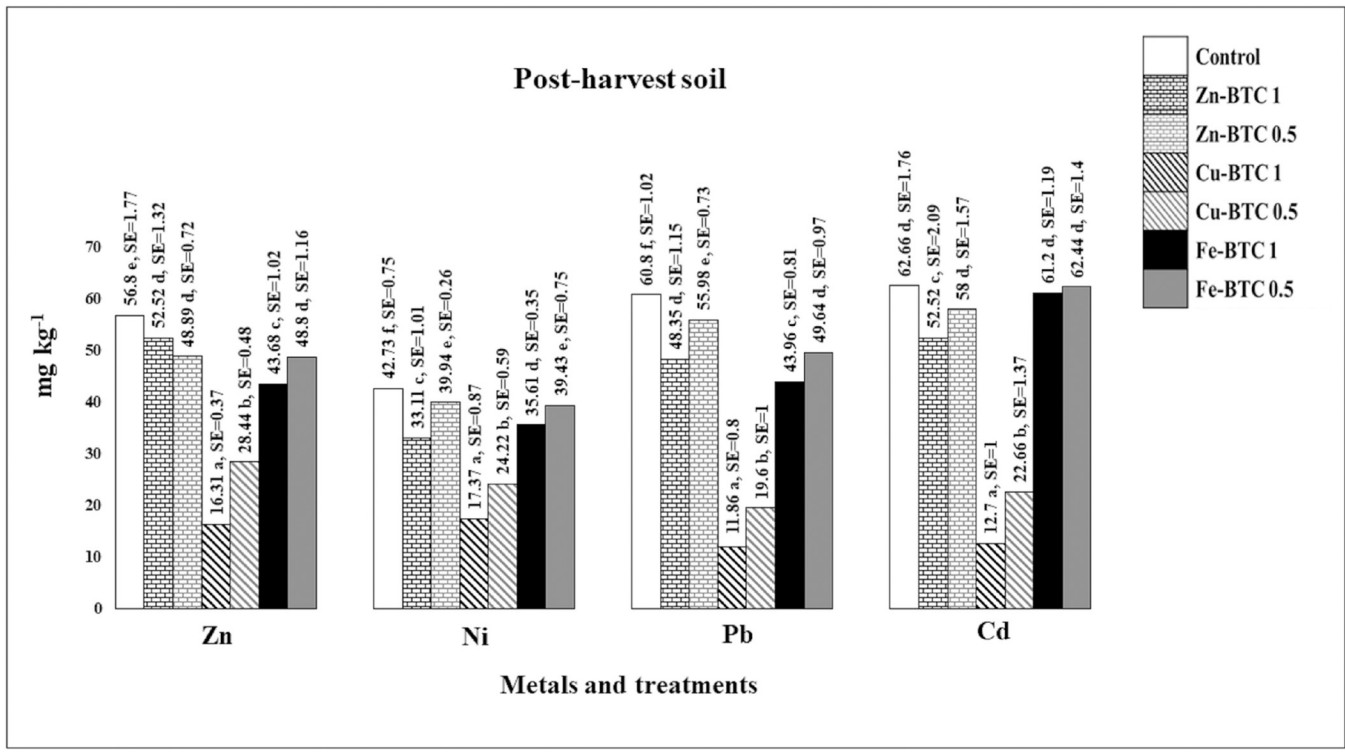

**Fig 6. Effects of Zn-BTC, Cu-BTC, and Fe-BTC on the extracted HMs by EDTA in the post-harvest soils (mgkg$^{-1}$).** Compare mean analysis have been done for each metal separately. Values with the same letters indicate non-significant difference at P < 0.05 according to the Duncan's multiple range test. Each value is the average of three replications. SE: Std. Error.

which was 52.52 followed by 48.89 mgkg$^{-1}$ under the effect of Zn-BTC 0.5%. EDTA was able to extract HMs from soil with great strength due to its acidic property. Gupta and Sinha [63] reported that EDTA, due to its acidic pH, can extract metals from unavailable fractions of soil in addition to bioavailable fraction. The lowest concentrations of Zn extracted by EDTA were observed in pots under the effect of Cu-BTC 1% and then 0.5%, which were 16.31 and 28.44 mgkg$^{-1}$, respectively. Both values were significantly different from the control treatment. The concentrations of Zn extracted by EDTA under the effects of 1 and 0.5% of Fe-BTC were 43.68 and 48.8 mgkg$^{-1}$, respectively, and both values showed a significant difference in comparison with the control. The lowest concentrations of Ni extracted by EDTA occurred under the effects of 1 and 0.5% of the Cu-BTC (17.37 and 24.22 mgkg$^{-1}$, respectively), which were significantly different from the control. After that, the lowest EDTA extractable Ni (33.11 mgkg$^{-1}$) was observed under the effect of Zn-BTC 1%, which was significantly different from the control. The differences between 1 and 0.5% rates of all three MOFs were significant for the EDTA extractable Ni. The lowest extracted Pb was observed significantly different from the control following the application of Cu-BTC 1 and 0.5%, which were 11.86 and 19.6 mgkg$^{-1}$, respectively. For Cd, like the previous HMs, the most significant decrease occurred in pots under the effects of 1 and 0.5% of the Cu-BTC, including 12.7 and 22.66 mgkg$^{-1}$, respectively. Zn-BTC 0.5%, Fe-BTC 1%, and Fe-BTC 0.5% treatments showed non-significant effects on reducing the extracted Cd compared to the control, which were 58, 61.2, and 62.44 mgkg$^{-1}$, respectively. In general, it was concluded that the Cu-BTC treatments showed the most remarkable effects on reducing the values of bioavailable fraction of all four HMs. It can be resulted that the Cu-BTC stabilized HMs in the soil more strongly than the other synthesized MOFs. Considering the acidity of the EDTA extractant, the lowest concentration of the extracted metals in pots amended with the Cu-BTC was as a reason for the greater structural strength of this MOF compared to the other synthesized MOFs. In addition, the most metal stabilization by the Cu-BTC was observed for Pb followed by Cd. In previous studies, EDTA has been reported as a strong complexing agent for HMs, especially Pb and Cd [64, 65]. Similarly, in the Cu-BTC treatment, the lowest metal uptakes by the plant shoot and root were observed for Pb followed by Cd. The highest affinity of Pb and Cd compared to other metals with Cu-based MOFs in aqueous solution has been reported by Shi et al. [66] and Wang et al. [67].

Due to intense industrial activities in the urban zones of Iran, the potential ecological risk of Pb and Cd has been reported to be very high (PER >150). Therefore, the agricultural soils of these areas may also be affected by high levels of these HMs [68]. Therefore, the synthesized Cu-BTC with a porous structure can have a very high efficiency on immobilizing these HMs in the soil. The ability of the synthesized MOFs to stabilize HMs in the studied soil was also related to their structural characteristics similar to the interpretation mentioned for the plant analysis.

In a soil contaminated with HMs with a concentration of less than 100 mgkg$^{-1}$, the concentrations of Ni, Pb, and Cd in old quinoa leaves were reported to be 2.99, 44.63, and 59.76 mgkg$^{-1}$, respectively [69]. While in the present study, with a concentration of 100 mgkg$^{-1}$ of each HM, the concentrations of Ni, Pb, and Cd in shoots were observed to be 31.83, 16.72, and 21.14 mgkg$^{-1}$ in the Cu-BTC (0.5%) treatment.

After the Cu-BTC, Zn-BTC and Fe-BTC showed the highest stabilization power of soil HMs. The process of the MOFs synthesis and their influences on the stabilization of the HMs and vegetative growth of quinoa is shown in Fig 7.

## Potential limitations of the study and suggestions

One of the limiting resources in the present study was the need for large Teflon autoclaves to synthesize MOFs in large quantities for addition to pots. Another issue was the need for a long

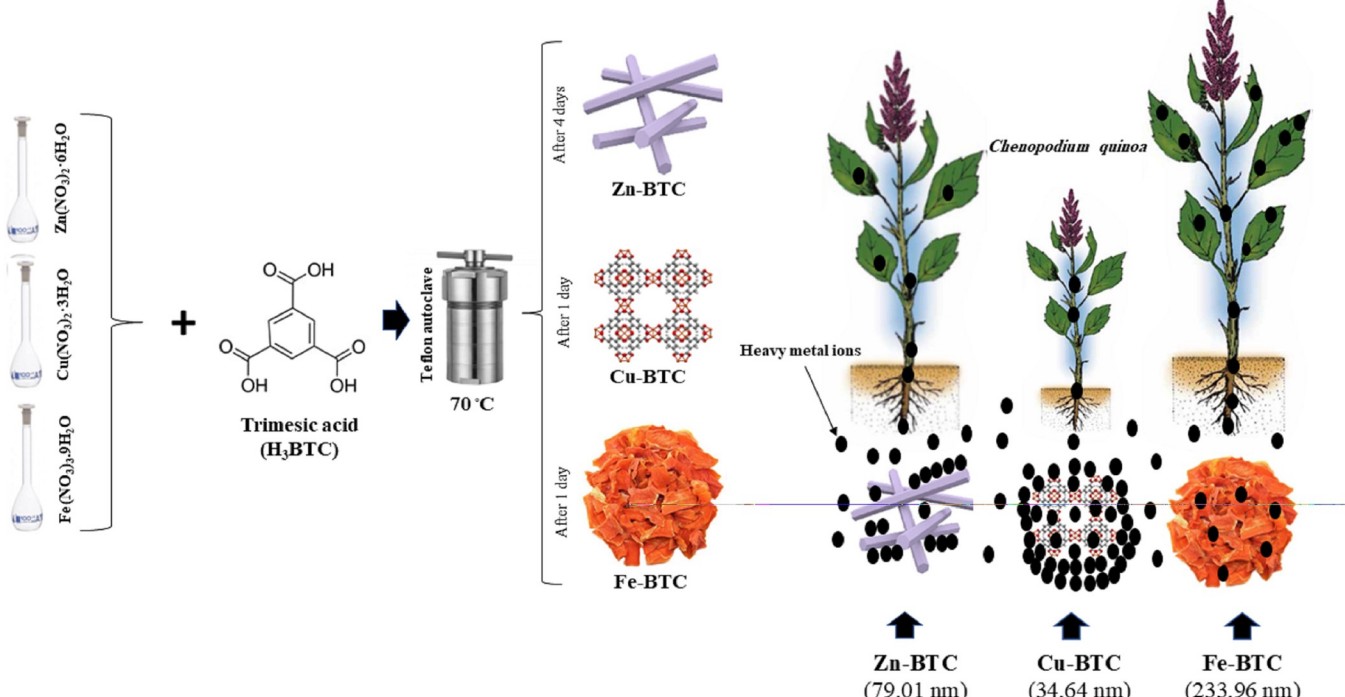

**Fig 7. Schematic representation of the Zn-BTC, Cu-BTC, and Fe-BTC synthesis and their effects on the stabilization of HMs and vegetative growth of quinoa.**

time (four days) to synthesize Zn-BTC to complete cristalization of this MOF. But the mentioned limiting factors were not an obstacle for conducting the present study.

On the other hand, it was observed that monometallic MOFs were highly effcient in stabilizing HMs in soil. The synthesis of these MOFs is faster and easier than MOFs consisting two or more metals. Meanwhile, Cu ions showed unique properties. The highly porous and honeycomb structure of MOFs synthesized by Cu ions has significantly increased the efficiency of this MOF compared to the other two MOFs. Therefore, the use of Cu-BTC even at levels lower than 1% is suggested to remove HMs in contaminated soils. Because the bioavailable form of HMs is very high in acidic soils, the toxicity of these metals is also high in these soils. Therefore, the application of Cu-MOFs in acidic contaminated soils is highly recommended. In addition, the use of other ligands other than trimesic acid can be suggested. For example, oxalic acid and malonic acid are suitable with their simple and simmetrical structures.

## Conclusion

The results of this research showed that the synthesized metal-$H_3BTC$ MOFs had a significant efficiency to remove HMs from the contaminated soil. Zn-BTC, Cu-BTC, and Fe-BTC as three metal-$H_3BTC$ MOFs with diverse structural properties were synthesized in the present study. The Fe-BTC had a semi-gel and flake-shaped and the Zn-BTC had a crystalline and rod-shaped structure. The Fe-BTC had the lowest specific surface area because of its semi-amorphous state. The synthesized Cu-BTC showed the strongest HMs stabilizition properties due to having the highest specific surface area, the most crystalline structure in the shape of a honeycomb, and the smallest structure size. The results showed that the lowest uptakes of HMs by the shoots and roots of quinoa cultivated in a multi-metal contaminated soil and the concentrations of HMs extracted with EDTA in the post-harvest soils were observed in the Cu-BTC

amended pots. After the Cu-BTC, the Zn-BTC significantly reduced the HMs uptakes and the highest uptake by the plant happened in the Fe-BTC amended pots. The lowest MPI indices were observed in shoot and root under the influence of Cu-BTC 1%. Among the synthesized MOFs and the HMs in the studied soil, the highest affinity was observed between Pb and the Cu-BTC. It was found that the MOFs with a porous structure had a much higher efficiency than other non-porous MOFs on immobilizing HMs in the soil. Therefore, MOFs based on Cu can have high environmental efficiency. The application of the Cu-BTC in the urban areas with high pollution level of Pb can be a suitable environmental solution to remove this metal from contaminated soils.

## Supporting information

**S1 Table. Data related to the uptake of HMs by shoot.**
(DOCX)

**S2 Table. Data related to the uptake of HMs by root.**
(DOCX)

**S3 Table. Data related to HMs extracted by EDTA.**
(DOCX)

**S1 File. Data related to XRD analysis.**
(DOCX)

## Acknowledgments

The authors are grateful to the Department of Soil Science, School of Agriculture, Shiraz University, Shiraz, Iran for providing research facilities.

## Author Contributions

**Conceptualization:** Amir Zarrabi, Reza Ghasemi-Fasaei, Abdolmajid Ronaghi, Sedigheh Zeinali, Sedigheh Safarzadeh.

**Data curation:** Amir Zarrabi, Reza Ghasemi-Fasaei, Abdolmajid Ronaghi, Sedigheh Zeinali, Sedigheh Safarzadeh.

**Formal analysis:** Amir Zarrabi, Reza Ghasemi-Fasaei.

**Funding acquisition:** Reza Ghasemi-Fasaei.

**Investigation:** Amir Zarrabi, Reza Ghasemi-Fasaei, Abdolmajid Ronaghi, Sedigheh Zeinali, Sedigheh Safarzadeh.

**Methodology:** Amir Zarrabi, Reza Ghasemi-Fasaei.

**Project administration:** Reza Ghasemi-Fasaei.

**Resources:** Amir Zarrabi, Abdolmajid Ronaghi.

**Software:** Amir Zarrabi.

**Supervision:** Reza Ghasemi-Fasaei, Abdolmajid Ronaghi, Sedigheh Zeinali, Sedigheh Safarzadeh.

**Validation:** Abdolmajid Ronaghi, Sedigheh Zeinali, Sedigheh Safarzadeh.

**Visualization:** Amir Zarrabi.

**Writing – original draft:** Amir Zarrabi, Abdolmajid Ronaghi.

**Writing – review & editing:** Amir Zarrabi, Reza Ghasemi-Fasaei.

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
