## [Decision Letter · Decision Letter 0]

24 Jun 2024

PONE-D-24-18022Application of the synthesized metal-trimesic acid frameworks for the remediation of a multi-metal polluted soil: Investigation of crop responsesPLOS ONE

Dear Dr. Ghasemi-Fasaei,

Thank you for submitting your manuscript to PLOS ONE. After careful consideration, we feel that it has merit but does not fully meet PLOS ONE’s publication criteria as it currently stands. Therefore, we invite you to submit a revised version of the manuscript that addresses the points raised during the review process.

We look forward to receiving your revised manuscript.

Kind regards,

Zhongchuang Liu, Ph.D.

Academic Editor

PLOS ONE

Journal Requirements:

Additional Editor Comments :

Please refer to the following article during the revision process.

Liu, Z.C., Wang, L.A., Zeng, F.T., Al-Hamadani, S.M.Z.F., 2015. The Absorption and Enrichment Condition of Mercury by Three Plant Species. Polish Journal of Environmental Studies. 24, 887-891.

Liu, Z.C., Wang, L.A., Xu, T.T., Deng, X.J., Zhang, H.L., 2014. Research on the effect of Na2S2O3 on mercury transfer ability of two plant species. Ecological Engineering. 73, 649-652.

Reviewers' comments:

Reviewer's Responses to Questions

**Comments to the Author**

1. Is the manuscript technically sound, and do the data support the conclusions?

Reviewer #1: Yes

Reviewer #2: Yes

Reviewer #3: Yes

2. Has the statistical analysis been performed appropriately and rigorously? 

Reviewer #1: N/A

Reviewer #2: Yes

Reviewer #3: Yes

3. Have the authors made all data underlying the findings in their manuscript fully available?

Reviewer #1: No

Reviewer #2: Yes

Reviewer #3: Yes

4. Is the manuscript presented in an intelligible fashion and written in standard English?

Reviewer #1: Yes

Reviewer #2: Yes

Reviewer #3: No

5. Review Comments to the Author

Reviewer #1: Considerably improve the quality of images and captions to convey the message clearly (self-sufficient) to the reader without relying on support from the text.

Observe the writing of scientific names of plants in italics, for example:

Chenopodium quinoa

Nicotiana tabacum

Morus alba

Davidia involucrata

Reviewer #2: Title

The title is clear and accurately reflects the content and scope of the study. It effectively highlights the use of metal-trimesic acid frameworks for soil remediation and the investigation of crop responses.

Abstract

The abstract provides a concise summary of the study, including the background, methodology, key results, and conclusions. However, it lacks a clear statement of the study's objectives at the beginning and does not sufficiently highlight the significance and implications of the findings for environmental remediation practices.

Introduction

The study provides a comprehensive background on the issue of heavy metal contamination in soils and clearly states the importance of using MOFs for soil remediation. It also introduces quinoa as a relevant crop for studying metal uptake. However, the introduction could benefit from more recent references to support the claims and provide a broader context. The rationale for choosing quinoa as the test crop could be further elaborated, particularly in the context of its properties relevant to heavy metal uptake.

Materials and methods

The statistical analysis section lacks detail on the methods used for data analysis including specific tests used and software employed. Some procedural steps could be described with more clarity to ensure replicability.

Results and discussion

The section does not adequately address potential limitations of the study, such as any assumptions made, limitations of the experimental setup, and potential sources of error. It also lacks suggestions for future research directions to build on the findings, such as exploring different types of MOFs or testing in different soil conditions.

Conclusion

Include a brief statement on the broader implications of the study's findings for environmental remediation practices, particularly in urban areas with high levels of heavy metal pollution.

References

Update the reference list to include more recent studies, particularly those published in the last five years, to provide a broader context and support for the study's findings.

General Comments

The manuscript presents a well-conducted study on the application of metal-trimesic acid frameworks for soil remediation. The findings are significant and contribute to the field of environmental science. With some minor revisions and additional explanations, the manuscript will be suitable for publication in PLOS ONE.

I recommend the manuscript for publication after the authors address the specific and general comments outlined above.

Reviewer #3: 1. Suggested Topic:

Application of the synthesized metal-trimesic acid frameworks for the

remediation of a multi-metal polluted soil and investigation on Quinoa responses.

2. Abstract, Results etc. :

Authors should replace "belongs to" with 'is for' or any other appropriate replacement

3. Authors should define abbreviation at the first use to aid comprehension, e.g. FTIR

4. It may be good to benchmark the HMs present in Quinoa in this research with the daily acceptable dosage from literature.

5. Authors have done a good first draft of reporting this interesting study however, further corrections have to be done. Some prepositions are missing and some sentences need restructuring. Editing tools should be employed to correct syntax, semantics and grammatical errors.

6. PLOS authors have the option to publish the peer review history of their article (what does this mean?). If published, this will include your full peer review and any attached files.

Reviewer #1: No

Reviewer #2: No

Reviewer #3: No

---

## [Author Response · Author response to Decision Letter 0]

22 Jul 2024

Dear professor Zhongchuang Liu

Academic Editor of PLOS ONE 

Thank you very much for considering our article.

The revisions have been done according to what you and reviewers asked.

*******

Dear professor Zhongchuang Liu

1. The manuscript has met PLOS ONE's style requirements, including those for file naming and the PLOS ONE style templates. 

2. The captions for Supporting Information files were included at the end of our manuscript according to Supporting Information guidelines. 

Your suggested articles have been used in the present article in the sections of Introduction and Materials and methods and they have been marked with green font color in the text.

Dear Reviewer #1

• The quality of the images has been adjusted by PLOS ONE Pace Image Tool. However, the length and width of the images have been increased again as much as possible.

• The scientific names of plants have been written in italics and marked in light orange font color in the text.

Dear Reviewer #2

Your recommended revisions have been marked with purple font color in the text: 

• The study's objectives at the beginning, the significance and implications of the findings for environmental remediation practices have been more clearly stated in the Abstract section.

• The reason for choosing quinoa for this study is given in the Introduction section. Old references have been replaced with new references (2024) in this section.

• Statistical analysis is described in more detail in the Materials and Methods section.

• The limitations and problems of conducting the experiment are explained under the heading of Potential limitations of the study and suggestions in the Results and discussion section. The significant role of copper in the production of high-performance MOFs is also mentioned. The use of copper MOFs in acidic contaminated soils has been mentioned. Also, the use of other organic ligands to synthesis MOFs is suggested for future studies.

• In the conclusion section, the high efficiency of copper MOF is mentioned due to its unique structure, especially for the removal of Pb in polluted urban soils.

• A number of new references (2024) have been replaced with old references or added in References section.

Dear Reviewer #3

Your recommended revisions have been marked with light blue font color in the text:

• The previous title of the article has been changed according to your recommendation.

• "belongs to" has been replaced by other appropriate words throughout the text.

• Heavy metal absorption by quinoa in this study has been compared to other studies.

• Many grammatical corrections have been made in the text, marked by Track Changes.

Best regards

Dr. Reza Ghasemi-Fasaei

Corresponding author

---

## [Decision Letter · Decision Letter 1]

7 Aug 2024

PONE-D-24-18022R1Application of synthesized metal-trimesic acid frameworks for the remediation of a multi-metal polluted soil and investigation of quinoa responsesPLOS ONE

Dear Dr. Ghasemi-Fasaei,

Thank you for submitting your manuscript to PLOS ONE. After careful consideration, we feel that it has merit but does not fully meet PLOS ONE’s publication criteria as it currently stands. Therefore, we invite you to submit a revised version of the manuscript that addresses the points raised during the review process.

We look forward to receiving your revised manuscript.

Kind regards,

Zhongchuang Liu, Ph.D.

Academic Editor

PLOS ONE

Journal Requirements:

**Additional Editor Comments:**

The plant names that need to be italicized in the references have not been modified.

The name of the vertical axis in Fig.1 is unclear.

Reviewers' comments:

Reviewer's Responses to Questions

**Comments to the Author**

1. If the authors have adequately addressed your comments raised in a previous round of review and you feel that this manuscript is now acceptable for publication, you may indicate that here to bypass the “Comments to the Author” section, enter your conflict of interest statement in the “Confidential to Editor” section, and submit your "Accept" recommendation.

Reviewer #3: All comments have been addressed

2. Is the manuscript technically sound, and do the data support the conclusions?

Reviewer #3: (No Response)

3. Has the statistical analysis been performed appropriately and rigorously? 

Reviewer #3: (No Response)

4. Have the authors made all data underlying the findings in their manuscript fully available?

Reviewer #3: (No Response)

5. Is the manuscript presented in an intelligible fashion and written in standard English?

Reviewer #3: (No Response)

6. Review Comments to the Author

Reviewer #3: (No Response)

7. PLOS authors have the option to publish the peer review history of their article (what does this mean?). If published, this will include your full peer review and any attached files.

Reviewer #3: No

---

## [Author Response · Author response to Decision Letter 1]

15 Aug 2024

Dear professor Zhongchuang Liu

Academic Editor of PLOS ONE 

Thank you very much again for considering our article.

The second round of revisions have been done according to what you recommended.

*******

Dear professor Zhongchuang Liu

• There was a retracted reference that was replaced by another reference. In reference number 35, Haseeb et al. 2022 was replaced by Bamagoos et al. 2022.

• The scientific names of plants that were not written in Italic in the references section were corrected.

• Fig 1 was edited and the name of the vertical axis is now clear.

Best regards

Dr. Reza Ghasemi-Fasaei

Corresponding author

---

## [Editor Report · Decision Letter 2]

19 Aug 2024

Application of synthesized metal-trimesic acid frameworks for the

remediation of a multi-metal polluted soil and investigation of quinoa responses

PONE-D-24-18022R2

Dear Dr. Ghasemi-Fasaei,

We’re pleased to inform you that your manuscript has been judged scientifically suitable for publication and will be formally accepted for publication once it meets all outstanding technical requirements.

Kind regards,

Zhongchuang Liu, Ph.D.

Academic Editor

PLOS ONE
---

## [Editor Report · Acceptance letter]

27 Aug 2024

PONE-D-24-18022R2 

PLOS ONE

Dear Dr. Ghasemi-Fasaei, 

I'm pleased to inform you that your manuscript has been deemed suitable for publication in PLOS ONE. Congratulations! Your manuscript is now being handed over to our production team.

Kind regards, 

on behalf of

Dr. Zhongchuang Liu 

Academic Editor

PLOS ONE